# Allosteric regulation of kinase activity in living cells

Shivani Sujay Godbole[1], Nikolay V Dokholyan[1,2,3,4,5]*

[1]Department of Pharmacology, Penn State College of Medicine, Hershey, United States; [2]Department of Biomedical Engineering, Penn State University, University Park, Hershey, United States; [3]Department of Engineering Science and Mechanics, Penn State University, University Park, Hershey, United States; [4]Department of Biochemistry & Molecular Biology, Penn State College of Medicine, Hershey, United States; [5]Department of Chemistry, Penn State University, University Park, Hershey, United States

*For correspondence:
dokh@psu.edu

Competing interest: The authors declare that no competing interests exist.

**Abstract** The dysregulation of protein kinases is associated with multiple diseases due to the kinases' involvement in a variety of cell signaling pathways. Manipulating protein kinase function, by controlling the active site, is a promising therapeutic and investigative strategy to mitigate and study diseases. Kinase active sites share structural similarities, making it difficult to specifically target one kinase, and allosteric control allows specific regulation and study of kinase function without directly targeting the active site. Allosteric sites are distal to the active site but coupled via a dynamic network of inter-atomic interactions between residues in the protein. Establishing an allosteric control over a kinase requires understanding the allosteric wiring of the protein. Computational techniques offer effective and inexpensive mapping of the allosteric sites on a protein. Here, we discuss the methods to map and regulate allosteric communications in proteins, and strategies to establish control over kinase functions in live cells and organisms. Protein molecules, or 'sensors,' are engineered to function as tools to control allosteric activity of the protein as these sensors have high spatiotemporal resolution and help in understanding cell phenotypes after immediate activation or inactivation of a kinase. Traditional methods used to study protein functions, such as knockout, knockdown, or mutation, cannot offer a sufficiently high spatiotemporal resolution. We discuss the modern repertoire of tools to regulate protein kinases as we enter a new era in deciphering cellular signaling and developing novel approaches to treat diseases associated with signal dysregulation.

## eLife assessment

One of the most promising strategies in development of drugs targeting kinases is provided by using allosteric control that allows specific regulation and study of kinase function without directly targeting the active site. This **important** paper reviews **convincingly** the current repertoire of tools for regulating the activity of protein kinases with the ultimate goal of developing novel approaches in treating diseases associated with signal dysregulation.

## Introduction

Protein kinases expedite the cell signaling processes by facilitating the transfer of a phosphate from the nucleotide triphosphate to the target protein. Kinase-mediated phosphorylation is critical for a wide spectrum of cell signaling processes such as transcription, translation, cell proliferation, receptor downregulation, metabolism, and apoptosis (*Cheng et al., 2011*). The dysregulation of kinases alters cell function and may lead to diseases such as schizophrenia (*Bentea et al., 2019*), Alzheimer's disease

**Figure 1.** Domains of kinase protein responsible for the regulation of function.

(*Ohno, 2014*), cancer (*Sarkar et al., 2023*), and cardiovascular diseases (*Chen et al., 2022a*). Kinases undergo dynamic conformational changes to elicit their function during cell signaling (*Dixon et al., 2004*). All kinases have a conserved residue sequence that forms the nucleotide-binding domain to bind to adenosine triphosphate (ATP) or guanosine triphosphate (GTP) (*Cheng et al., 2011*; *Taylor and Kornev, 2011*). The binding of the nucleotides is facilitated by phosphorylation of the activation loop (*Adams, 2003*). Therapies targeting kinases focus on the nucleotide-binding sites (active/orthosteric sites) that regulate the kinase function. Therapies targeting orthosteric sites to regulate kinase activity compete with the native ligand (ATP or GTP) to bind to the active site and need to be highly selective for the active site. Due to the conservation of the nucleotide-binding domain across all protein kinases, specifically targeting the active/orthosteric sites of one type of kinase is challenging.

Kinase activity is also regulated by autoinhibitory domains (*Soderling, 1990*; *Li et al., 2021*). Autoinhibitory domains suppress the interaction between the kinases and the effector molecule (nucleotides). The autoinhibitory domains are the sites distal to the active sites that affect the activity of the kinase at the active site through the induction of conformational and dynamic changes in the protein structure (*Nussinov and Tsai, 2013*; *Cui and Karplus, 2008*; *Guo and Zhou, 2016*). Allosteric regulation does not involve direct competition with the native ligand. Instead, it perturbs active site structure or dynamics. The kinases, especially in diseased conditions, may develop resistance to orthosterically regulated therapies by incorporating mutations at the active site, thus enabling kinases to elicit catalytic activity while disallowing the binding of the orthosteric regulators. Allosteric control of kinase function allows the bypass of this resistance (*Cui and Karplus, 2008*).

Finding allosteric sites that effectively regulate active sites is critical for establishing allosteric control over a particular kinase. X-ray crystallography (*Keedy, 2019*) and nuclear magnetic resonance (NMR) spectroscopy (*Grutsch et al., 2016*) techniques are widely used to observe the structures of a protein. The changes occurring in the structure of protein molecules during the allosteric binding of effector molecules can be mapped by comparing structures at different points in time. However, these techniques are expensive and time-consuming. Modern computational approaches offer an inexpensive and fast assessment of protein allosteric 'wiring.' However, the computational approaches only offer hypothetical predictions and still need to be tested experimentally. Experimental validation does not require the structural characterization of the protein; instead, a direct test of the predictions in activity assays demonstrates the functioning of the established allosteric control (*Vishweshwaraiah et al.,*

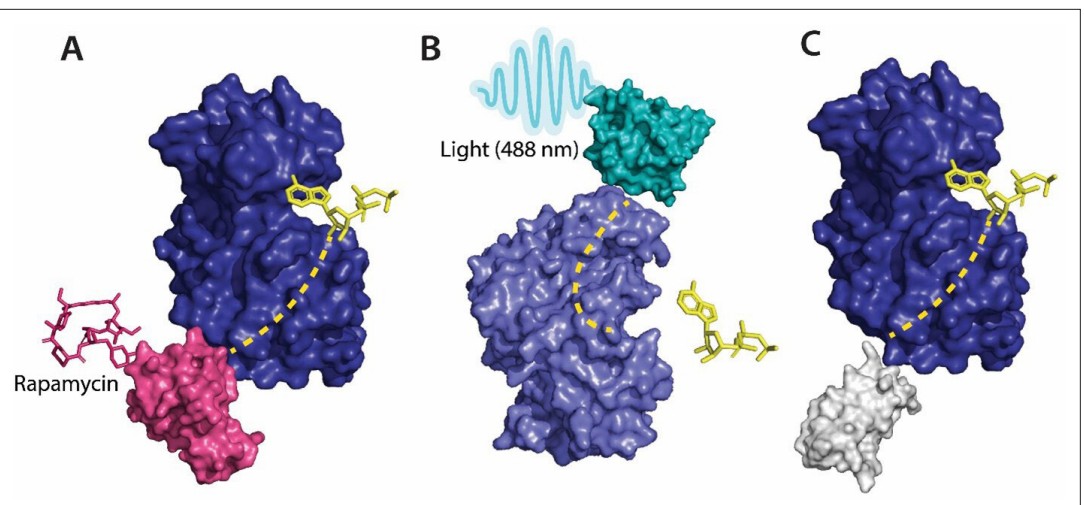

**Figure 2.** Various tools used as allosteric regulators. (**A**) Allosteric activation of kinase (ERK2 kinase, PDB ID: 4GT3) by uniRapR (PDB ID: 7F2J) domain that binds to small molecule (rapamycin) causing activation of the protein. (**B**) Allosteric inhibition of kinase (ERK2 kinase, PDB ID: 4GT3) by insertion of an optogenetic control protein, LOV2 (PDB ID: 2V0W), causing conformational change upon irradiation by blue light. (**C**) Activation of kinase using monobodies (PDB ID: 3RZW).

The online version of this article includes the following source data for figure 2:

**Source data 1.** Source data for *Figure 2*.

*2021b*). Here, we discuss computational and experimental approaches to map allosteric communications, and tools used to control protein function through allosteric regulation.

## Allosteric regulation of protein kinases

Kinases are considered to be biological switches due to their ability to change structural conformation upon activation (*Taylor et al., 2012b*). Since kinases make up 2% of the protein-coding genome (*Manning et al., 2002*) in humans and are implicated in multiple diseases, this family of proteins is an important therapeutic target. Protein kinases phosphorylate serines, threonines, or tyrosines by transferring the γ-phosphate of ATP to the hydroxyl group causing structural changes in the protein, eliciting its function as an enzyme (*Cohen, 2002*). Kinases have a widespread role in cellular signaling, thereby making them desirable targets for establishing control over cellular processes (*Vishweshwaraiah et al., 2021b*; *Chen et al., 2022b*). In eukaryotic cells, across all different types of protein kinases, there are close to 250 residues worthy of conserved sequences (*Taylor et al., 2012b*). One conserved region is attributed to be the nucleotide-binding domain (NBD), which is a cleft that forms the binding pocket for ATP and is located between the N-terminal lobe and the C-terminal lobe of the protein. The flexibility of the protein is maintained by two ensembles of hydrophobic residues or 'spines' that connect the α-helix of the C lobe to the β-sheet of the N lobe (*Johnson and Lewis, 2001*). These spines are called the 'regulatory spine (R spine)' due to their ability to regulate the assembly and disassembly of the active kinase, and the 'catalytic spine (C spine)' since these residues balance the catalytic activity of the kinases upon ATP binding. The changes in structure of the kinases due to their complex structure–activity relationship are likely to be resolved due to the presence of the spines (*Taylor et al., 2012a*). The dynamic structure of kinases has proved to be of great importance for the functionality of the protein and the NMR structures confirming these dynamic structural changes have underscored the presence and importance of allostery of protein kinases (*Masterson et al., 2012*; *Figure 1*).

Several approaches have been developed using the preexisting tools to control protein function for the allosteric regulation of protein kinases. Kinase activity can be controlled by ligands (*Zorba et al., 2019*), made use of monobodies (*Sha et al., 2017*) (a type of synthetic binding protein) to control allostery in Aurora A (AurA) (*Zhou et al., 1998*). AurA is a protein kinase responsible for the maturation of the centrosome and duplication through its effect on the microtubule spindle formation

in mitosis (*Glover et al., 1995*). AurA serves to be a major oncogene since its overexpression has been associated with a variety of cancers. The authors used monobodies (*Koide et al., 2012*) to bind to the allosteric sites of AurA and produce complete inhibition or complete activation without having to actively compete with ATP to bind to the NBD. The effect of the binding of monobodies to AurA and the activity of the protein were mapped using X-ray crystallography. The binding of the monobody showed a 20-fold inhibition of the kinases, proving its effectiveness as an allosteric inhibitor and demonstrating promising therapeutic potential. *Wojcik et al., 2016* designed high-affinity monobodies to bind to the kinase Bcr-Abl formed by the fusion of breakpoint cluster region (Bcr) and Abelson tyrosine kinase 1 (Abl1). Bcr-Abl activity leads to the development of chronic myelogenous leukemia (*Deininger et al., 2000*). Studies have shown an interaction between Src homology domain (SH2) and tyrosine kinase, critical to kinase activity and mutations arising at these interfaces (I164E), and leads to inhibition of the kinase activity and oncogenesis induced by Bcr-Abl (*Grebien et al., 2011*). The authors developed monobodies for these critical residues to control the kinase activity of Bcr-Abl allosterically. The monobodies were developed using combinatorial synthesis and the binding affinity was tested using chemical shift perturbation assays that identify binding sites and predict the binding efficiency of ligands. Inhibition of interaction of the SH2 domain and the kinase domain due to monobodies decreased the activity of Bcr-Abl (validated by western blot and immunohistochemistry experiments). Monobodies can be used as a tool for controlling kinase activity, providing an alternative approach to chemogenetic and optogenetic control of protein function as monobodies do not require engineering of the target kinase (*Figure 2*).

*Dagliyan et al., 2017* developed a rapamycin-binding domain that destabilized the NBD, thereby disrupting the kinase activity in the P21-activated kinase (PAK1). PAK1 is known to play a role in angiogenesis, neurogenesis, and cell motility (*Hofmann et al., 2004*). The dysregulation of this protein leads to the development of speech impairment, seizures, and macrocephaly. The authors developed a protein domain that is activated upon dopamine binding to the PAK1 domain. PAK1 is overexpressed in breast cancer and hence is identified as an oncogene (*Radu et al., 2014*). The authors tested this rapamycin-PAK construct (RapR-PAK1) in breast cancer cells to observe changes in the kinase activity due to the effect of rapamycin. The construct was successfully able to control the kinase activity of PAK, demonstrating the applicability of ligands to control protein activity allosterically by either inhibiting activity or inducing it.

*Karginov et al., 2011* used a similar strategy for regulating kinase activity, but instead of using a ligand-based approach, the authors used light as the input signal for controlling activity. The authors built a photoactivatable analog of rapamycin to regulate the dimerization of proteins. The photo-inducible small molecules are generated by the installation of 'caging groups' (*Riggsbee and Deiters, 2010*; *Mayer and Heckel, 2006*) that keep the molecule inactive until removed. Caging groups sequester the site on the ligand necessary for activation. Dissociation of the caging groups from the activation site through irradiation by light leads to activation of the small molecules. The authors used 'caged-rapamycin (pRap)' to regulate both the kinase activity and protein dimerization.

*Zhou et al., 2017* developed an optically inducible, single-chain protein that could undergo conformational changes upon exposure to light. The authors induced modifications to DronpaN145, a photo-inducible green fluorescent protein to form pdDronpa1 (*Zhou et al., 2012*) that could dimerize at the kinase active site in the presence of 500 nm cyan and stymie kinase activity. Upon removal of light, the kinase activity was restored. The authors used dual-specificity mitogen-activated protein kinase 1 (MEK1) to test the pdDronpa1 construct. The authors were able to successfully control the kinase activity of MEK1, providing a unique and versatile approach toward optogenetics.

*Vishweshwaraiah et al., 2021a* developed a protein-based nanocomputing agent (*Dokholyan, 2021*) containing two inputs that had opposing effects on the kinase activity, forming a logic OR gate to regulate allosteric communications. The authors inserted a chemogenetic (uniRapR; *Dagliyan et al., 2013*) and an optogenetic domain (LOV2; *Lee et al., 2008*) within FAK. The uniRapR construct was inserted in the allosteric site of the kinase domain, predicted using Ohm (*Wang et al., 2020*), such that it could activate the protein upon interaction with rapamycin. This activation pathway of the protein upon interaction with rapamycin was predicted by discrete molecular dynamics (DMD) simulations. The LOV2 construct was inserted in loop 2 of the FERM domain (allosteric site on FERM domain was predicted using Ohm), leading to an inhibition of kinase activity by inactivation of the protein (*Lietha et al., 2007*). Inactivation of the protein was predicted using DMD simulations in blue light on/

off conditions. The authors observed a decrease in kinase activity upon exposure to blue light and an increase in kinase activity upon exposure to rapamycin upon live cell imaging (changes in phenotype were used to determine kinase activity). This successful study provides a basis for the development of rationally designed nano-computing agents and can be used in diagnostics, drug delivery, regulation of cell signaling, and metabolic flux. In order to demonstrate that the nanocomputing agents are effective in other proteins as well, Chen et al. developed an Src-uniRapR-LOV2 construct that provided a dual control on the function of Src kinase. The uniRapR construct was able to activate the Src kinase while the LOV2 construct rendered the protein inactive upon irradiation by blue light. The on/off protein switch was tested by visualization of the focal adhesion. With rapamycin as the first signal, followed by blue light as the second (*Chen et al., 2023*). Along with computational methods such as multiple sequence alignments (MSA), molecular dynamics (MD), and DMD simulations, molecular docking and in silico modeling (*Yuan et al., 2022*). paired with mass spectroscopy, have also been used to define protein structures and map allosteric sites in proteins. Through microscopic techniques, such as cryo-EM, two-dimensional structures of proteins can be acquired and morphed into three-dimensional models using artificial intelligence (*Chirigati, 2021*). In addition to small molecules, biologics such as antibodies or various input stimuli (such as pH, temperature, or pressure) can be made use of to regulate protein function by exploitation of allosteric communication. Protein allostery proves to be an essential tool in controlling protein function in translational research.

## Methods to map allosteric communications

Computer simulations developed to predict allosteric communications offer faster and more cost-effective methods of studying allosteric communication in proteins compared to traditional experimental structural characterization techniques (*Dokholyan, 2016*). Experimentally, NMR spectroscopy and X-ray crystallography are used in conjunction to map allostery, *Boulton and Melacini, 2016* hypothesized that the crystal structures of proteins solved by these techniques would provide information about the changes in the conformation of proteins upon binding of an allosteric regulator. Sites that are susceptible to a conformational change upon binding of the modulator/regulator were termed 'hot spots.' The high-resolution images from NMR spectroscopy provided the opportunity to study the conformational dynamics as a result of allostery. By combining the high-resolution images of the protein structure and the MD simulations, predictions can be made about the allosteric communications within proteins. *Lisi and Loria, 2017* described allostery using NMR studies paired with the computational predictions in enzyme catalysts; PTP1B (protein tyrosine phosphatase 1B) (*Zhang, 2017*), which is known to be associated with obesity and diabetes, and PKA-C (protein kinase C) (*Xiao et al., 2015*), which is responsible for cell signaling and proliferation. *Jain et al., 2023* used NMR spectroscopy to configure the active and inactive structures of Cdc42-LOV2 construct. Light oxygen voltage (LOV2) protein is used as an optogenetic tool since it is active under blue light and is used to control protein function allosterically. The authors elucidated the structural changes of the Cdc42-LOV2 occurring upon irradiation by blue light converting the construct from an inactive to an active state. This provides an insight on the allosteric communication within proteins through structural elucidation. NMR spectroscopy is able to map both structural and dynamic changes that occur within an allosteric system, but may not be sufficient to determine the structural and dynamic changes occurring in complex allosteric systems (*Boulton and Melacini, 2016*).

Computational methods provide a faster means of deciphering protein interactions. *Gadiyaram et al., 2019* developed a graph spectral analysis of protein structures to predict allostery using non-covalent interactions between amino acid residues to create a protein structure network. The authors put their theories to the test by predicting the side-chain interactions between residues that form the G-protein coupled receptors (GPCRs) (specifically the β2-adrenergic receptor) and the HIV proteases. Another computational strategy, MSA (*Sofi and Masoodi, 2022*), is the process of aligning two or more protein sequences to discover the conserved regions of the proteins. *Lockless and Ranganathan, 1999* made use of MSA to map allosteric communications in a postsynaptic density protein, *Drosophila* disc large tumor suppressor, and zonula occludens-1 (PDZ) protein domain. They hypothesized that perturbations in one residue of the protein sequence would cause thermodynamic changes in the residue adjacent to it. If the protein sequences are conserved, similar changes in the vibrational frequencies should be seen in all proteins possessing the conserved residue sequence. The theory had two developmental parameters – conservation of the residues at a given site of the

protein as observed in the MSA, and statistical coupling of two residues (nodes) of the given amino acid sequence. This method was validated by analyzing the conserved sequences of the PDZ and the Poxvirus and zinc finger (POZ) protein domains using statistical coupling analysis (*Teşileanu et al., 2015*) and thermodynamic mutation cycles (*Pagano et al., 2021*), which measures the strength of the interaction of the residues. *Ambroggio and Kuhlman, 2006* developed an algorithm to engineer proteins that can undergo structural protein-folding changes using MSA. The algorithm Ambroggio and Kulhman developed could predict stable structural conformations of both the changed and native protein instead of one protein conformation favoring the other. To test the algorithm, the authors made use of the Sw2 sequence, which could switch the conformations between the coiled coil and the zinc finger states. The information provided by MSA is susceptible to errors (*Kemena and Notredame, 2009*) as the 'true alignment' of proteins is not possible limiting the accuracy of allosteric communication predictions using MSA (*Nuin et al., 2006*).

The calculations necessary to describe protein structure flexibility and allostery of large molecular systems are computationally intensive. MD simulations help to overcome this limitation in predicting the interactions between amino acid residues. MD simulations use atomic interaction approximations to simulate atomic motions on the basis of Newtonian physics and reduce the computational complexity of this systems (*Durrant and McCammon, 2011*). MD simulations are utilized to develop algorithms that predict allosteric communications. The algorithms not only help in predicting allosteric communication pathways within proteins but also help in determining the potential allosteric sites in the protein. *Proctor et al., 2015* analyzed the allosteric communications in mutations of cystic fibrosis transmembrane conductance regulator protein (CFTR) and identified the allosteric hot spots using molecular dynamic simulations. These hot spots could be targeted to restore the transcription and translation of CFTR and provide an allosteric therapeutic approach. MD simulations can predict residue interactions within proteins at a high spatiotemporal resolution and provide a faster method of analysis of allosteric communication within a protein compared to experimental methods used for structure elucidation. The timescales play a role in predicting residue interactions since allosteric communications are dynamic in nature. Short time scales (nanosecond) are not always accurately predicted by MD simulations (*Hertig et al., 2016*); this can be overcome using coarse-grained simulations (a type of multiscale modeling method that simulates complex behavior of system in a more realistic timescale) or DMD simulations that reach time scales in the range of microseconds (*Shirvanyants et al., 2012*; *Proctor et al., 2011*). DMD simulations have been successfully used to predict allosteric communications in proteins (*Bowerman and Wereszczynski, 2016*; *Dokholyan et al., 1998*).

*Tee et al., 2021* developed a structure-based statistical mechanical model of allostery (SBSMMA). This model predicts allostery by quantifying the thermodynamic perturbations of proteins (upon ligand binding or mutations in the protein) by estimating the free energy of every residue present in the ligand-bound and ligand-free proteins using an allosteric signaling map (ASM). SBSMMA identifies potential exosites, protein regions that are likely to govern the allosteric regulation through reverse perturbations. AlloSigMA (*Guarnera et al., 2017*) is a server that provides a quantitative assessment of protein-energy dynamics by estimating the changes in the relative energy of the residues at the allosteric site. AlloMAPS (*Tan et al., 2019*) is a database that provides data on the predicted energy changes and the allosteric pathways of the molecule predicted using SBSMMA (*Tee et al., 2021*). SBSMMA provides a way to discover allosteric sites on the protein and elucidate allosteric mechanisms within proteins. *Wah Tan et al., 2022* developed a protocol for designing allosteric modulators using a fragment-based approach based on the knowledge of allosteric interaction sites on a protein using SBSMMA. Fragment-based approach (*Scott et al., 2012*) of drug design is the development of an entire small molecule by joining smaller, low molecular-mass fragments that show similar interactions with the target. This method was developed by *Jencks, 1981* and was initially used for building small molecules for kinases. Fragment-based approach can be used to map protein–protein interactions, transcription factors, and protein chaperons (*Chen et al., 2010*).

Treating the allosteric phenomenon as signal propagation through heterogeneous media, *Wang et al., 2020* designed a physics-based computational approach for mapping allosteric communications called Ohm (*Wang et al., 2020*). The multifaceted platform can be used to predict the allosteric sites along with the possible pathways, identify the critical residues, and predict the possible allosteric correlation between the residues. The platform is built on a perturbation propagation algorithm that measures the perturbation of residues by calculating the probability of interactions between the two

residues (obtained by a probability matrix). The predictions were validated by predicting the allosteric communication during phosphorylation of CheY (*Lee et al., 2001*) (a protein found in the flagellar motor of bacteria) and the Caspase-I protein (*Clark, 2016*) (involved in inflammatory and host-defense response). Allosteric communication within proteins can be predicted by a variety of computational methods that provide closely realistic predictions in a short span of time. These predictions can be validated using experimental techniques.

## Tools to control allostery

Conceptually, a 'sensor' domain is engineered into an allosteric site to establish allosteric control over a protein. This sensor domain responds to stimuli, external or internal, and undergoes conformational changes that propagate through allosteric paths to the active site. The active site then is altered upon the application of the stimuli. Some of the examples are where 'sensor' proteins interact with small molecules or light and produce a conformational and dynamic change in the target protein leading to either activation or deactivation of the protein.

## Optogenetic control

Photoreceptors are molecules that can change their conformation upon irradiation by light. The light-sensitive domains of the photoreceptors can be used as sensor domains to control protein function (*Padmanabhan et al., 2022*). The phytochrome B (PhyB) (*Ni et al., 1999*) is used extensively as a red light photoreceptor since many optical systems do not have far-infrared light range lasers, and red light photoreceptors have limitations to their usage (*Ni et al., 1999*). Light oxygen voltage (LOV2) and cryptochromes are the two main types of blue light photoreceptors used to control protein function. The (LOV2) domain of phototropin 1 (*Halavaty and Moffat, 2007*) obtained from *Avena sativa* has been shown to produce dramatic conformational changes upon exposure to blue light and is used to control protein function. *Zimmerman et al., 2016* developed LOV2 photoswitches and validated them by inserting them in various protein domains and determining the activity of the protein upon exposure to blue light. *Dagliyan et al., 2016* developed a photo-inhibitable Src kinase domain by inserting LOV2 in the L1 loop connecting β-strands of Src kinase causing inhibition of the kinase upon activation of LOV2. However, insertion of the LOV2 domain into the protein without causing structural perturbations in the native protein structure is difficult due to limitations arising from the lack of availability of caging surfaces leading to non-specific activation of the photoreceptors. Caging surfaces or photocages are photosensitive groups that prevent the protein from reacting to small molecules or other proteins that might produce a biological function of the protein. The caging groups ensure activation of proteins only upon irradiation of the proteins by light of suitable wavelength (*Bojtár et al., 2020*). To overcome this limitation, *He et al., 2021* developed a circular permutant of LOV2 (cpLOV2). The cpLOV2 is constructed by introduction of a glycine and serine-rich linker in the surface-exposed loop. The construct remains compatible with the existing LOV2-based engineered proteins with the advantage of having more caging surfaces thus improving its applicability (*Figure 2*).

Owing to the difficulty of optogenetic tools to control allosteric activity, arising due to the limitations in achieving subcellular control of activity (*Wang et al., 2019*, *Shaaya et al., 2020*) developed an optogenetic tool, LightR, that enables subcellular control of protein and provides a tight temporal regulation of protein activity. LightR comprises of two tandemly connected Vivid photoreceptor (*Nihongaki et al., 2014*) domains that form an antiparallel homodimer upon irradiation by blue light causing its activation. The use of LightR in controlling allosteric activity was validated in Src kinase, where irradiation with blue light caused activation of Src kinase. LightR provides better spatiotemporal resolution than LOV2 due to the subcellular control of proteins (*Figure 3*).

## Chemogenetic control

Although optogenetics provides a high resolution of control of protein activity, optogenetic tools are often challenging to translate from cellular to animal models due to the limitation of the extent of absorption of light by animals that are non-transparent. To overcome this limitation, *Dagliyan et al., 2013* developed a protein switch (uniRapR) whose activity can be controlled by a small molecule (rapamycin). uniRapR is a complex of FK506-binding protein (FKBP12) and FKBP12-rapamycin binding protein (*Choi et al., 1996*) built into a single-chain protein. uniRapR binds selectively to rapamycin and upon binding induces a conformational change in the structure. uniRapR can be used to control

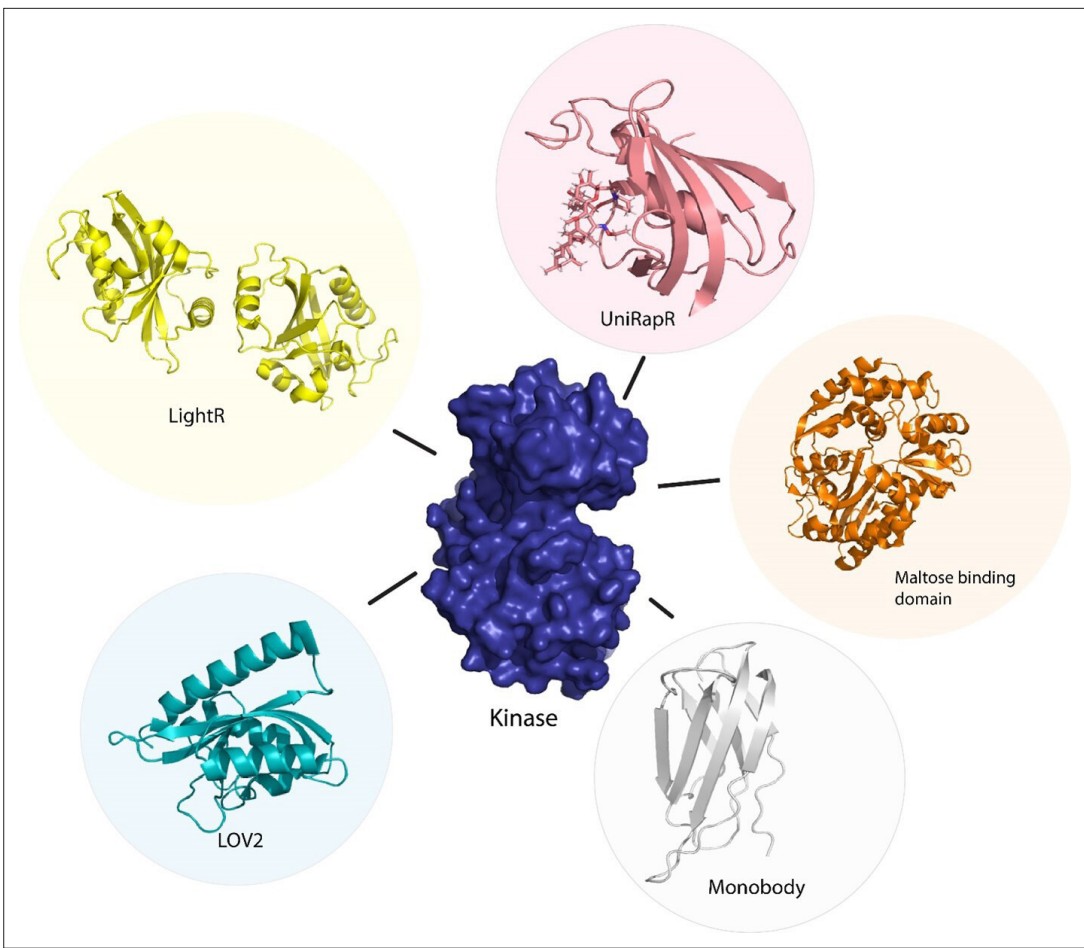

**Figure 3.** Sensor proteins used to control function of kinase protein allosterically. Chemogenetic tools (LightR, LOV2), optogenetic tool (uniRapR, maltose binding protein), and synthetic protein tool (monobody).

protein function allosterically. To test the functionality of uniRapR, the authors inserted this construct into Src kinase and focal adhesion kinase (FAK) and compared the changes in phosphorylation of the two enzymes before and after the addition of rapamycin. The activity of uniRapR was further validated using a mutant strain of FAK with a point mutation that caused inhibition of FAK activity; the FAK activity was recovered by uniRapR, which validated its use as an allosteric protein function regulator.

*Guntas and Ostermeier, 2004* developed a maltose-binding allosteric regulator using in vitro recombination techniques. The wild-type beta-lactamase enzyme (BLA) activity is rendered unaffected by *Escherichia coli* maltose binding protein. Upon recombination of the two genes, TEM-1-beta-lactamase and *E. coli*. maltose binding gene (*Guntas and Ostermeier, 2004*), the enzyme activity was seen to be regulated upon the addition of maltose. The change in activity was analyzed by measuring the rate of β-lactam hydrolysis upon an increase in the concentration of maltose. Chemogenetic control of proteins has better in vivo resolution since light can only be absorbed up to a certain depth and has a shorter timescale of activation (*Figure 2*).

## Conclusion

The human cell kinome consists of 538 kinases that are involved in a variety of cellular processes. Kinases have a conserved active site residue sequence, making it challenging to target a specific kinase with ligands acting on the active site; hence, allosteric regulation of kinases provides the ability to specifically target different types of kinases. Finding the allosteric sites on the proteins and visualizing the changes occurring in the protein conformation is experimentally time-consuming and an expensive endeavor. To overcome this limitation, computational methods are developed that can predict allosteric communication within proteins. Advancements in technology have paved the way to

engineer proteins and regulate their activity. Various methods such as chemogenetics, optogenetics, and synthetic proteins are used as allosteric tools to control protein function. Kinases are an important family of proteins as they are implicated in a variety of diseases; hence, allosteric regulation of kinases could be used as a potential therapeutic approach. Combining the various allosteric control tools helps develop nanocomputing agents with more than one input function, leading to a different output function of the protein, providing a means of controlling different aspects of protein function. New tools can be developed that react to different inputs such as temperature, pH, and pressure along with the preexisting tools interacting with light or small molecules. These allosteric control sensor proteins provide a better spatiotemporal resolution in controlling protein function and prove to be a potential therapeutic approach targeting diseases that arise from the dysregulation of proteins.

## Acknowledgements

We thank Dr. Richard Mailman, Brianna Hnath, and Jiaxing Chen for valuable discussions. We acknowledge support from the National Institutes for Health (R35 GM134864, 1RF1 AG071675-01, and 1R01 AT012053) and the Passan Foundation.

## Additional information

### Funding

| Funder | Grant reference number | Author |
|---|---|---|
| National Institutes of Health | R35 GM134864 | Nikolay V Dokholyan |
| National Institutes of Health | 1RF1 AG071675 | Nikolay V Dokholyan |
| National Institutes of Health | 1R01 AT012053 | Nikolay V Dokholyan |

The funders had no role in study design, data collection and interpretation, or the decision to submit the work for publication.

### Author contributions

Shivani Sujay Godbole, Nikolay V Dokholyan, Conceptualization, Writing – original draft

### Author ORCIDs

Shivani Sujay Godbole (iD) http://orcid.org/0000-0002-0177-2286
Nikolay V Dokholyan (iD) https://orcid.org/0000-0002-8225-4025

Joint Public Review: https://doi.org/10.7554/eLife.90574.4.sa1
Author Response https://doi.org/10.7554/eLife.90574.4.sa2

## Additional files

### Supplementary files
• MDAR checklist

### Data availability
The accession codes deposited in PDB for the protein structures used in Figure 2 (4GT3, 2V0W, 3RZW) are included in the manuscript.

The following previously published datasets were used:

| Author(s) | Year | Dataset title | Dataset URL | Database and Identifier |
|---|---|---|---|---|
| Pozharski E, Zhang J, Shapiro P | 2012 | ATP-bound form of the ERK2 kinase | https://www.rcsb.org/structure/4GT3 | RCSB Protein Data Bank, 4GT3 |
| Halavaty AS, Moffat K | 2007 | N- and C-terminal helices of oat LOV2 (404-546) are involved in light- induced signal transduction (cryo-trapped light structure of LOV2 (404-546)) | https://www.rcsb.org/structure/2V0W | RCSB Protein Data Bank, 2V0W |
| Gilbreth RN, Koide S | 2011 | Crystal Structure of the Monobody ySMB-9 bound to human SUMO1 | https://www.rcsb.org/structure/3RZW | RCSB Protein Data Bank, 3RZW |

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
