## [Editor Report · eLife assessment]

One of the most promising strategies in development of drugs targeting kinases is provided by using allosteric control that allows specific regulation and study of kinase function without directly targeting the active site. This **important** paper reviews **convincingly** the current repertoire of tools for regulating the activity of protein kinases with the ultimate goal of developing novel approaches in treating diseases associated with signal dysregulation.

---

## [Referee Report · Joint Public Review]

This concise review provides a clear and instructive picture of the state-of-the-art understanding of protein kinases' activity and sets of approaches and tools to analyse and regulate it.

Three major parts of the work include: methods to map allosteric communications, tools to control allostery, and allosteric regulation of protein kinases. The work provides an important and timely view of the current status of our understanding of the function of protein kinases and state-of-the-art methods to study its allosteric regulation and to develop allosteric approaches to control it.

---

## [Author Response]

The following is the authors’ response to the previous reviews

**Comments from reviewer 1:**
Comment 1. Regarding SBSMMA, the authors may complement their discussion by mentioning recent work (PMID: 35738428) where SBSMMA was used to exemplify a potential fragment-based design approach for developing allosteric effectors for kinases.

Thank you for the suggestion, we have added a short summary of the work where SBSMMA is used as a basis for developing small molecules to target kinases using fragment-based design approach